# Chronic Exposure of Renal Progenitor Cells (HRTPT) to As (III) Implicates Microfibril Associated Protein 5 (MFAP5) in the Activation of Carcinoembryonic Antigen Related Cell Adhesion Molecules (CEACAM 5 and 6)

**DOI:** 10.3390/cimb47060455

**Published:** 2025-06-12

**Authors:** Md Ehsanul Haque, Donald A. Sens, Scott H. Garrett

**Affiliations:** Department of Pathology, School of Medicine and Health Sciences, University of North Dakota, Grand Forks, ND 58202, USAdonald.sens@und.edu (D.A.S.)

**Keywords:** arsenite (As III), HRTPT, renal fibrosis, MFAP5, CEACAM 5 and 6

## Abstract

Studies on populations exposed to inorganic arsenic (iAs) have shown an association with the development of chronic kidney disease (CKD) and renal cell carcinoma (RCC). However, there are few studies addressing how acute exposure of the human kidney to iAs might lead to the long-term alterations that might lead to CKD or RCC. This laboratory’s hypothesis is that renal exposure to iAs might alter the renal cells responsible for the repair and regeneration of nephrons damaged by iAs exposure or other renal toxicants. The kidney possesses a minority epithelial cell population that co-expresses PROM1 and CD24, which are believed to be involved in renal epithelial cell repair. The purpose of this work is to understand the pathogenesis of CKD in renal cortical epithelial cells. Our model consists of acute and chronic exposure of i-As (III) to “Human Renal Tubular Precursor TERT” (HRTPT). The microarray and gene validation study demonstrated a sudden induction of microfibril associated protein 5 (MFAP5) and carcinoembryonic antigen related cell adhesion molecule 5 and 6 (CEACAM 5 and 6) in chronic i-As (III)-exposed cells. Chronically exposed cells also exhibited an induction of the pAKT/AKT pathway and SOX9 transcription factor. The targeting of MFAP5 and CEACAM 5/6 could, therefore, provide a potential therapeutic approach to CKD.

## 1. Introduction

Anthropogenic activities throughout the world have resulted in substantial contamination of the earth’s air, soil, and water with heavy metals. As a class, these elements present a serious health concern [1,2,3]. Arsenic is one of the most abundant of these elements, and its toxicity is one of the highest of all the heavy metal toxicants. Epidemiological studies have shown that it is carcinogenic to the skin, lungs, bladder, liver, and kidneys [4,5,6,7,8,9]. In the United States alone, over 2.1 million people are believed to be drinking arsenic-contaminated domestic well water and from other sources, both known and unknown [10]. In the U.S., arsenic levels in drinking water cannot exceed 10 µg/L, as recommended by the World Health Organization and enforced by the US Environmental Protection Agency [11,12,13]. Research has shown that arsenic metabolism occurs primarily in the liver; recent studies have shown that arsenic is also metabolized and eliminated by the human kidney’s proximal tubules (PT) [14,15,16]. These tubules serve as the locations for the energy-intensive reabsorption of nutrients, sodium chloride, sodium bicarbonate, and water (60–70%) [17]. Thus, the proximal tubule of the human kidney can sustain a number of injuries during the reabsorption and metabolism of xenobiotics from drugs and environmental toxins, including heavy metals. Arsenic accumulation in the proximal tubules can cause acute kidney injury (AKI), which progresses to chronic kidney disease (CKD) [4,18,19,20]. Moreover, according to epidemiological studies, CKD and kidney cancer are both associated with arsenic in drinking water [21,22,23,24,25]. Clear cell carcinoma (ccRCC) accounts for most renal cancers and is thought to arise from a cell within the PT [26]. Renal progenitor cells involved in the repair of renal damage are thought to arise from dedifferentiated PT or from scattered cells within the kidney with characteristics of PT [27,28,29,30]. Regardless of the origin of progenitor cells, only a few studies have examined the response of these cells to iAS [31]. In a previous study, a global gene expression analysis was performed between control and iAs-exposed HRTPT cells having co-expression of PROM1 and CD24 with progenitor cell properties [31,32]. An examination of the global gene expression profile showed the overexpression of carcinoembryonic antigen-related cell adhesion molecules (CEACAM 5). The goal of the present study was to validate this finding and to begin determining the mechanism underlying the overexpression of the CEACAM 5 genes.

## 2. Materials and Methods

### 2.1. Cell Culture and Reagents

Parental RPTEC-TERT cells were obtained from American Type Culture Collection (ATCC), and the HRTPT cell line has been previously described [33,34]. The culture medium consisted of a 1:1 mixture of DMEM:F12 serum-free media supplemented with selenium (5 ng/mL), insulin (5 μg/mL), transferrin (5 μg/mL), hydrocortisone (36 ng/mL), triiodothyronine (4 pg/mL), and epidermal growth factor (10 ng/mL) (Gibco, Waltham, MA, USA). Confluent cultures of HRTPT cells were exposed to 4.5 µM i-As (III) (III) (sodium arsenite) for 24 h and then sub-cultured at a 1:3 ratio in the continued presence of i-As (III) until confluent. Following confluence, the cells were serially sub-cultured again in the presence of i-As (III) until confluent. This was repeated for 19 serial passages. Every passage was examined microscopically, and harvested cells were preserved for RT-qPCR and protein analysis.

### 2.2. RNA Extraction and RT-qPCR

The mRNA and protein expression of individual genes was determined using RT qPCR and Western blotting as described previously [33,34,35]. Confluent cell cultures were harvested to obtain RNA cell pellets, which were then flash-frozen under liquid nitrogen. After lysing the cell pellets with 350 µL RLT^®^ buffer (Qiagen, Hilden, Germany), they were dissociated using QIA shredder tubes (Qiagen) for 2 min at 12,500 rpm. Isolation of RNA was carried out using QIAGEN’s RNeasy Mini Plus Kit (#74034) and QIAcube Instrument (Hilden, Germany), according to the manufacturer’s protocols. A NanoDrop spectrophotometer was used to quantify RNA (Thermo Fisher Scientific, Waltham, MA, USA). We synthesized cDNA from the total RNA using a LunaScript^®^ RT SuperMix Kit (New England Biolabs #E3010L, Ipswich, MA, USA) as recommended by the manufacturer. A final concentration of 20 ng/µL was obtained by diluting cDNA with nuclease-free water. Two microliters of cDNA (20 ng) was used for qPCR, and the results were analyzed using the BioRad CFX96 Touch Real-Time PCR Detection System (Hercules, CA, USA) and the Luna^®^ Universal qPCR Master Mix (New England Biolabs #M3003E). The parameters for the qPCR cycle were one 2-min cycle at 95 °C, forty 5-s cycles at 95 °C, and thirty seconds at 60 °C annealing temperature. Using 18S as the reference control gene, expression levels were calculated using threshold cycle (Ct) values using the 2^−∆∆Ct^ method.

### 2.3. Western Blot

Protein expression was determined by Western blot analysis using protocols that have been previously published by this laboratory [33,34,35,36]. The HRTPT cell pellets were lysed in an ice-cold RIPA buffer containing equal volumes of protease inhibitors, phenylmethylsulfonyl fluoride (PMSF), and sodium orthovanadate (Santa Cruz, CA, USA) and incubated 15 min on ice by shaking. The extracts were sheared twice using a sonicator for about 15 s each time, keeping them on ice, and centrifuged for 13,000× *g* for 10 min at 4 °C. The supernatants were transferred to fresh, cold microfuge tubes, and protein quantification was performed using the BCA assay (ThermoScientific, Waltham, MA, USA). After quantifying the samples, if required, each sample was diluted to 50 μg with RIPA, mixed with Laemmli buffer (Bio-Rad, Hercules, CA, USA), and boiled for 5 min at 95 °C. The samples were quickly centrifuged for 10 s and loaded on a TGX Any Kd^TM^ SDS polyacrylamide gel (Bio-Rad Laboratories, Hercules, CA, USA) and transferred to a nitrocellulose membrane (Amersham Biosciences, Piscataway, NJ, USA). The blots were blocked in tris-buffered saline (TBS) containing 0.1% Tween-20 (TBS-T) and 5% (*w*/*v*) non-fat dry milk for 1 h at room temperature on the same day after being rinsed for 5 min in TBST. Following blocking, the membranes underwent three 15-min TBS-T washes before being probed with the corresponding primary antibody for a whole night at 4 °C in a shaker. The primary antibodies were made using 3% BSA or 5% non-fat milk in TBS-T, and they were diluted using the proper dilution factor.

### 2.4. Statistical Analysis

The experiments were carried out in triplicate, and the data were analyzed using one-way ANOVA (non-parametric) with Tukey post hoc testing using GraphPad PRISM 10. Gene expression was normalized to the 18S housekeeping gene. The measurements were performed in triplicate for gene data. The reported values are mean ± SEM. A *t*-test was performed for mRNA and protein, respectively, and asterisks indicate significant differences from the control (* *p* < 0.05, ** *p* < 0.01, *** *p* < 0.001, **** *p* < 0.0001).

## 3. Results

### 3.1. Light Microscopic Evidence Shows Fibroblast-like Growth in Human Renal Progenitor Cells at Passage (P3) and (P8)

After exposure to 4.5 μM of As (III), HRTPT cell lines displayed morphological changes, including spindle-shaped and irregularly shaped cells. The cells undergo this transformation after several sequential passages with As (III), becoming irregularly shaped domes or lacking domes and exhibiting characteristics both similar to fibroblasts and more similar to mesenchymal cells (Figure 1). The domes form regularly in cells that have not been exposed to arsenite for prolonged periods. Overall, microscopic evidence indicates that renal progenitor cells exhibit fibroblast-like characteristics after being exposed to As (III).

### 3.2. Arsenite Exposure Increased the Expression of MFAP5, CEACAM 5, and CEACAM 6

MFAP5 mRNA levels and protein levels increased significantly with 4.5 µM As (III) exposure at P_3_ and P_8_ in HRTPT cells (Figure 2A–C). A similar result has been observed in cells exposed to As (III) that exclusively overexpress CEACAM 5 and 6 (Figure 3A,B). The Human Protein Atlas database shows that CEACAM 5 is expressed on the plasma membrane of cells (https://www.proteinatlas.org/ENSG00000105388-CEACAM5/subcellular, accessed on 27 March 2025), but the location of CEACAM 6 is unknown (Figure 3C). The CEACAM is a cell adhesion molecule related to carcinoembryonic antigen that functions as a cell surface receptor. Upregulation of MFAP5 and CEACAM 5 and 6 suggests a possible link to a pro-fibrotic phenotype. A microarray study conducted previously in our laboratory revealed a 2.4 K-fold increase in MFAP5 and a 0.4 K-fold increase in CEACAMs compared with the control at P3 (Appendix A, Table A1). This present study further validates the gene expression of both of them. In addition, we examined mRNA levels of fibrosis-related genes, such as fibronectin (FN1, 750-fold upregulation over control), and collagen (COL1A1, 9-fold increase over control) (Appendix B, Figure A1).

### 3.3. As (III) Exposure Increased the Expression of PI3K and MMP1

To gain a better understanding of the molecular cellular signaling pathways involved in cell growth, proliferation, invasion, and metastasis, the mRNA level of phosphoinositide 3-kinase (PI3K) and matrix metalloproteinase-1 (MMP1) was assessed in i-As (III)-exposed cells. A higher magnitude of PI3K and MMP expression was observed in cells exposed to arsenite (Figure 4). Western blot analysis shows elevated phopho-Akt in i-As (III)-exposed HRTPT cells compared to the unexposed control (Figure 5).

### 3.4. Determine How i-As (III) Increased the Expression of Phospho-Stress-Activated Protein Kinase/Jun N-Terminal Kinase (p-SAPK/JNK)

To gain a better understanding of the molecular cellular signaling pathways involved in renal fibrosis, the protein level of SAPK and p-SAPK was assessed in i-As (III)-exposed cells. A higher magnitude of p-SAPK expression was observed in cells exposed to arsenite. Western blot analysis shows elevated phopho-SAPK in i-As (III)-exposed HRTPT cells compared to the unexposed control (Figure 6).

### 3.5. Expression of Transcription Factors (TFs) in Cancer-Associated Fibroblast Activation

In order to determine transcription factors in cancer-associated fibroblast activation, which play a role in cell proliferation and cancer progression, we evaluated the expression of several transcription factors, including SOX9, POU5F1, ZEB1, JUN, and FOS in the HRTPT cell line exposed to arsenite (Figure 7). Chronically exposed cells also exhibited consistent induction of TFs; these markers are commonly activated in EMT pathways involved in CKD and or in RCC.

## 4. Discussion

According to the American Cancer Society, nine out of ten kidney cancers are renal cell carcinomas, while about seven out of ten people with RCC have clear cell renal cell carcinoma (ccRCC) [26]. Thus, RCC has become one of the greatest public health problems in the world, and American health care systems are forced to spend billions of dollars treating it. The recurrence of ccRCC occurs locally or distantly within five years after nephrectomy for 20–30% of patients. Cancer recurrence following surgery is one of the major factors negatively affecting patient survival. Using biomarkers, doctors and researchers can predict treatment outcomes, make informed treatment decisions, monitor cancer, and calculate the risk of recurrence [37,38,39,40]. It is, therefore, imperative to gain a better understanding of the pathogenesis and search for new biomarkers or therapeutic targets.

The present study focused on exploring how exposure to nephrotoxins might alter the renal epithelial cells charged with the repair and regeneration of the epithelial cells comprising a damaged nephron. For this study, the laboratory employed an immortalized human cell culture model, HRTPT, that co-expresses PROM1 and CD24 and displays the expected properties of a renal epithelial cell with capabilities for renal repair and regeneration [32]. In a previous study, the laboratory employed these cells to determine their response to i-As exposure [31]. It was shown that the HRTPT cells responded to i-As exposure by undergoing an epithelial-to-mesenchymal transition (EMT), which was mostly but not fully reversed upon removal of exposure of the cells to iAs [31]. As part of this study, global gene expression was performed on the HRTPT cells in the presence and absence of exposure to i-As. While the most attention was focused on the EMT and MET response of the HRTPT cells, it was also noted that the expressions of MFAP5 and CEACAM 5 were highly expressed due to i-As exposure. These genes and their regulation were chosen for validation and further study, since two recent studies suggest they might have a role in renal fibrosis and RCC [41,42,43,44]. One study showed that CEACAMs are prognostic markers for some malignancies, including ccRCC [42], while another study showed MFAP5 to be involved in cancer-associated fibroblast activation in pancreatic cancer [41]. The validation of these two markers showed a concurrent upregulation, which suggests a potential link to the development of renal fibrosis and cell transformation. This potential linkage was further explored through the analysis of several additional pathways likely to be involved in i-As-induced alterations in HRTPT cells. The i-As-exposed HRTPT cells showed high levels of mRNA for PI3K and p-AKT/AKT, which is interesting since PI3K activates signaling cascades that promote cell growth and division. The PI3K/AKT signaling pathway itself serves a major role in regulating cell physiology and pathology, including cell proliferation, survival, and invasion [45,46,47]. Other studies found that EMT and metastasis were promoted by overactive PI3K/AKT pathways because of their potent effects on cell migration [48,49,50]. The present study demonstrates that significant PI3K/AKT upregulation correlates with the many alterations in the microarray analysis associated with EMT and those associated with renal fibrosis.

The stress-activated protein kinase/Jun-amino-terminal kinase SAPK/JNK was also shown to be activated, which is interesting since it is activated by a variety of environmental stresses and inflammatory cytokines [51]. The activation of these pathways also correlates with the overexpression of matric components, such as FN1, COL1A1, ACTA2, and TAGLN, in i-As-exposed HRTPT cells (Appendix A and Appendix B). The expression of SOX9 was also examined since it is a transcription factor known for its role in promoting EMT and can bind to specific DNA sequences that regulate genes involved in cell phenotypic changes, migration, and invasion [52,53]. The HRTPT cells exposed to i-As upregulated factors such as POU5F1, Zeb1, JUN, and FOS at defined points within the time course of exposure to i-As.

Overall, these studies suggest a number of changes in the gene expression of progenitor-like cells when exposed to i-As. The present study is limited in scope due to several related factors. The HRTPT cells are immortalized, which presents advantages and disadvantages. An advantage is that renal fibrosis and renal cancer require long-term exposure to nephrotoxic agents. A disadvantage is the fact that they are immortalized, and this process itself may influence data interpretation. An additional advantage is that the HRTPT cells offer a longer term of exposure compared to primary cultures of renal cells that have a very limited lifespan along with difficulties in tissue acquisition. Thus, the present study largely provides insight into potential alterations in gene expression that might guide precise genomic analysis of intact human renal tissue by single-cell technology and spatial analysis. While the current results are descriptive, the data provides insight for further in vitro studies to identify alterations elicited by i-As on cells co-expressing both PROM1 and CD24. Gene knockdowns and knock-ins will be needed in future studies to begin to define cause and effect among the genes identified through global gene analysis.

## 5. Conclusions

Chronic kidney disease (CKD) and kidney cancer are both associated with arsenic in drinking water, according to epidemiological studies. Upon acute kidney injury, proximal tubular epithelial cells proliferate rapidly, which is due to their intrinsic proliferation potential. This remains a translational challenge to understand how progenitor cells regenerate after chronic nephrotoxic injury, despite substantial progress in understanding renal fibrosis and the role of progenitor cells. Most studies assume that insults and damage only occur to mature and differentiated functioning cells and not to regenerating progenitor cells. This is an issue that our system is uniquely positioned to address. An in vitro study carried out by our lab used human renal tubular progenitor cell lines to understand how chronic levels of i-As^+3^ induce EMT-like phenotypic and molecular changes. It is suggested that the concomitant upregulation of MFAP5 and CEACAM 5 and 6 is associated with renal fibrosis and cell transformation. It is anticipated that this study will contribute to the understanding and development of new therapeutic strategies to address RCC caused by chronic exposure to arsenic.

## Figures and Tables

**Figure 1 cimb-47-00455-f001:**
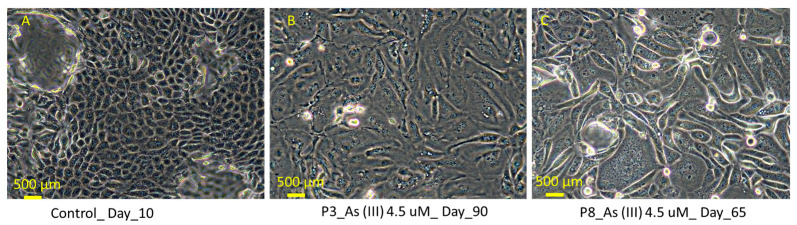
Light microscopic images of HRTPT cells exposed to 4.5 μM of arsenite show fibroblast-like growth at passage P_3_ (**B**) and P_8_ (**C**). (**A**) Control without arsenite; (**B**,**C**) exposed to arsenite passaged up to P_8_ and grown in 1:1 DMEM/F12 media.

**Figure 2 cimb-47-00455-f002:**
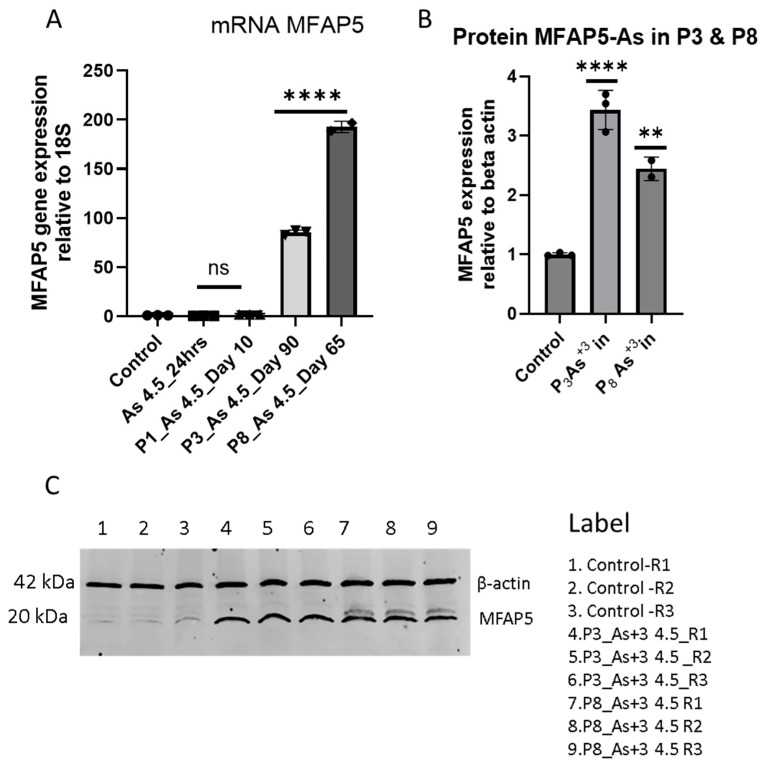
Expression of MFAP5 marker in HRTPT cell lines exposed to 4.5 µM of iAs up to P_8_ passages. RT-qPCR and Western blot analysis of (**A**) mRNA_MFPA5 and (**B**) protein_MFPA5 (**C**) Western blot. The expression of the MFAP5 gene and protein was normalized to the 18S housekeeping gene and β-actin, respectively. The measurements were performed in triplicate for gene and protein data. The reported values are mean ± SEM. A one-way ANOVA was performed, and **** and ** indicate significant differences in gene/protein expression level compared to the control 0.0 µM arsenite concentration at a *p*-value of ≤0.0001 and ≤0.01, respectively. R1, R2, and R3 correspond to replications used in control, P3, and P8, respectively. “ns” indicates non-significant.

**Figure 3 cimb-47-00455-f003:**
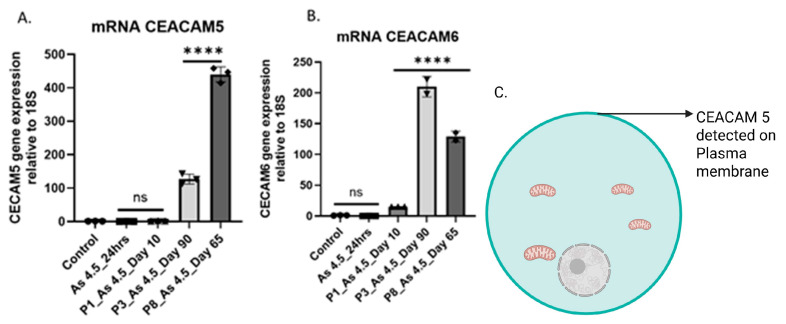
RT-qPCR analysis of CECAM 5 and 6. (**A**) CEACAM 5 and (**B**) CEACAM 6. **** indicate significant differences in mRNA level compared to the control 0.0 µM arsenite concentration at a *p*-value of ≤0.0001. (**C**) Expression of CEACAM 5 on the surface plasma membrane (light green color), but the location of CEACAM 6 is unknown. The expression of the CEACAMs was normalized to the 18S housekeeping gene. The measurements were performed in triplicate for gene data. The reported values are mean ± SEM. A one-way ANOVA was performed, and **** indicate significant differences in gene expression level compared to the control 0.0 µM arsenite concentration at a *p*-value of ≤0.0001. “ns” indicates non-significant.

**Figure 4 cimb-47-00455-f004:**
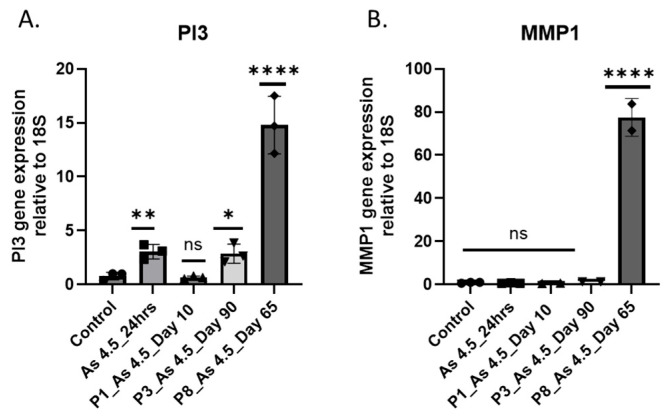
RT-qPCR analysis of PI3K and MMP1. (**A**) PI3K and (**B**) MMP1; the expression of the mRNA level was normalized to the 18S housekeeping gene. The measurements were performed in triplicate for gene and protein data. The reported values are mean ± SEM. A one-way ANOVA was performed, and ****; **; and * indicate significant differences in gene expression level compared to the control 0.0 µM arsenite concentration at a *p*-value of ≤0.0001; ≤0.01; and ≤0.05, respectively. “ns” indicates non-significant.

**Figure 5 cimb-47-00455-f005:**
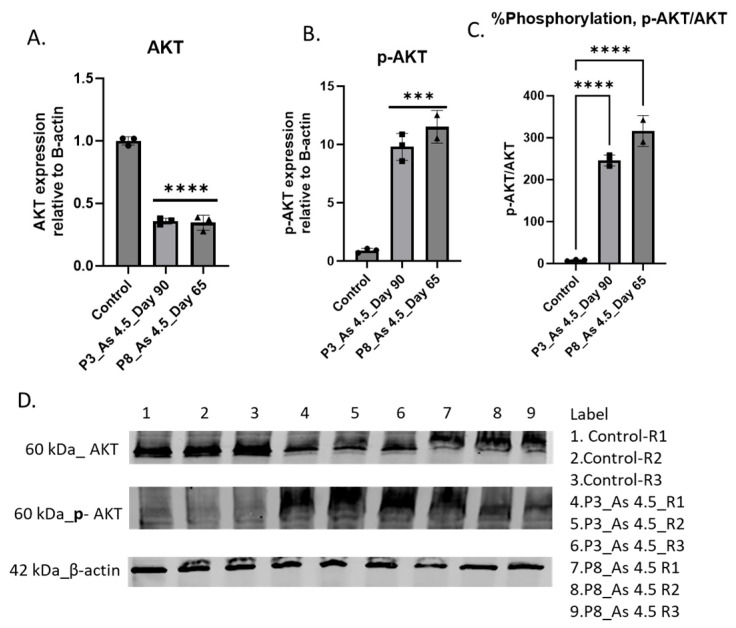
Western blot of Akt, phospho-Akt, and p-Akt/Akt in the HRTPT cell line exposed to arsenite. (**A**) Akt, (**B**) p-Akt, (**C**) p-Akt/Akt, (**D**) Western blot; **** and *** indicate significant differences in protein level compared to the control 0.0 µM arsenite concentration at a *p*-value of ≤0.0001; ≤0.001, respectively. R1, R2, and R3 correspond to replications used in control, P3, and P8, respectively. “ns” indicates non-significant.

**Figure 6 cimb-47-00455-f006:**
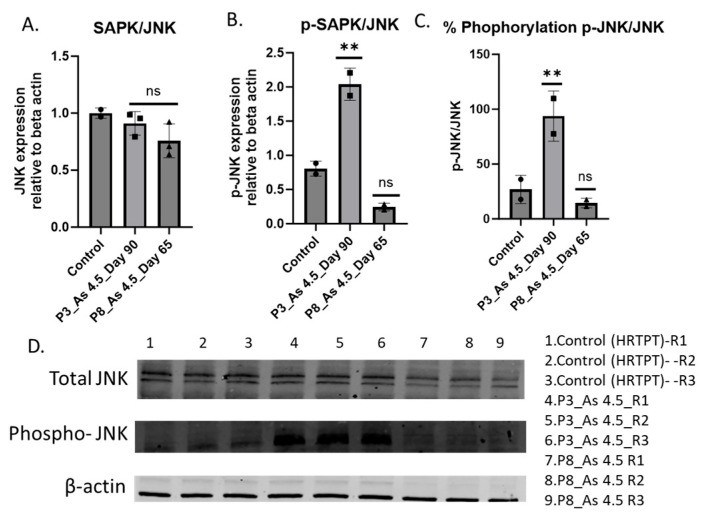
Western blot of SAPK, phospho-SAPK, and p-SAPK/SAPK in the HRTPT cell line exposed to arsenite. (**A**) Akt, (**B**) p-Akt, (**C**) p-Akt/Akt, (**D**) Western blot; ** indicate significant differences in protein level compared to the control 0.0 µM arsenite concentration at a ≤0.01. R1, R2, and R3 correspond to replications used in control, P3, and P8, respectively. “ns” indicates non-significant.

**Figure 7 cimb-47-00455-f007:**
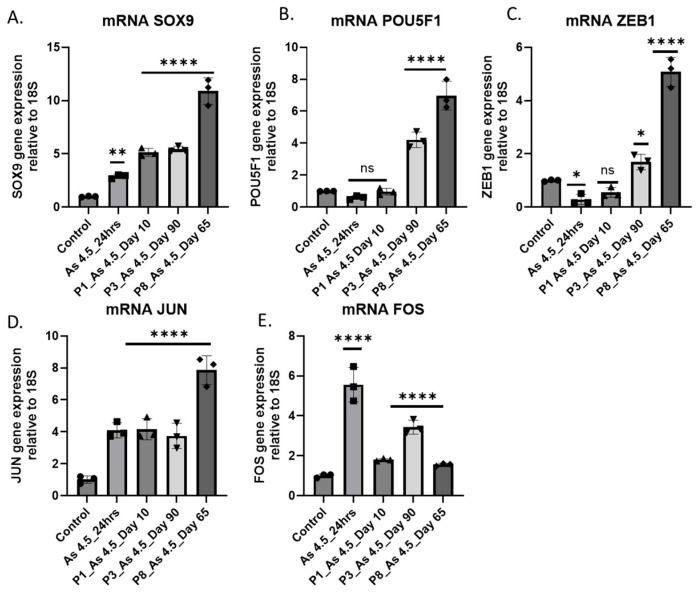
mRNA level of SOX9, POU5F1, ZEB1, JUN, and FOS in HRTPT cell line exposed to arsenite. (**A**) mRNA SOX9, (**B**) mRNA POU5F1, (**C**) mRNA ZEB1, (**D**) mRNA JUN, (**E**) mRNA FOS; ****; **; and * indicate significant differences in mRNA level compared to the control 0.0 µM arsenite concentration at a *p*-value of ≤0.0001; ≤0.01; and ≤0.05, respectively. “ns” indicates non-significant.

## Data Availability

The original contributions presented in this study are included in the article. Further inquiries can be directed to the corresponding author.

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
