# Peer review of "Chronic Exposure of Renal Progenitor Cells (HRTPT) to As (III) Implicates Microfibril Associated Protein 5 (MFAP5) in the Activation of Carcinoembryonic Antigen Related Cell Adhesion Molecules (CEACAM 5 and 6)"

_cimb, 2025, doi:10.3390/cimb47060455_

Round 1
Reviewer 1 Report
Comments and Suggestions for Authors
Revision of the manuscript biomedicines-3533574 entitled:
“Chronic Exposure of Renal Progenitor Cells (HRTPT) to As (III)
Implicates Microfibril-Associated Protein 5 (MFAP5) in the Activation of
Carcinoembryonic Antigen Related Cell Adhesion Molecules (CEACAM 5 & 6)”
Authors: MD EHSANUL HAQUE, Donald Sens, Scott Garrett
I am pleased to have the opportunity to review this interesting and well-executed paper, which offers valuable insights into the response of HRTPT cells to chronic exposure to As(III).
While the manuscript is of high quality overall, I have a few minor observations that should be addressed prior to publication:
- The paper is well-structured and generally clear. However, I strongly recommend including a graphical abstract. Given the extensive number of proteins and signaling pathways discussed, a visual summary would greatly enhance the clarity and accessibility of the study.
- The conclusions should be more clearly delineated. Presenting them in a concise and explicit manner would strengthen the overall impact of the manuscript.
- Please ensure that all abbreviations are spelled out in full upon their first appearance in the text. The majority of abbreviations currently lack full names.
- The Introduction, Materials and Methods, and Discussion sections are particularly well written and informative.
- In the Results section, specifically subsection 3.1, please provide clearer descriptions for P3 and P8. Additionally, there is some repetition in the figure legends for Figures 2 and 3 that should be revised for conciseness. In Figure 6, the Western blot results for phosphorylated JNK are not clearly visible. Enhancing the image quality would help support the findings.
Given the quality of the research and the relevance of the topic, I recommend acceptance of this manuscript following minor revisions. The study is well-conducted and will be of significant interest to both clinicians and researchers in nephrology and related fields.
Author Response
Reviewer 1.
I am pleased to have the opportunity to review this interesting and well-executed paper, which offers valuable insights into the response of HRTPT cells to chronic exposure to As(III).
While the manuscript is of high quality overall, I have a few minor observations that should be addressed prior to publication:
- The paper is well-structured and generally clear. However, I strongly recommend including a graphical abstract. Given the extensive number of proteins and signaling pathways discussed, a visual summary would greatly enhance the clarity and accessibility of the study.
Reply: a graphical abstract. (Attached World File, image enclosed there)
Summary
Microscopic evidence indicates that renal progenitor cells exhibited fibroblast & myofibroblast-like characteristics after being exposed to As (III) at P3
Molecular evidence shows induction of extracellular matrix (ECM) proteins, particularly Microfibril-Associated Protein 5 (MFAP5), suggestive of profibrotic processes, thus, it may play an important role in renal fibrosis.
The microarray and gene validation study demonstrated consistent upregulation of carcinoembryonic antigen-related cell adhesion molecule 5 & 6 (CEACAM 5& 6) in chronic i-As (III) exposed cells. Thus, CEACAM 5 & 6 (receptors)-MFAP5 (ligand) interaction activates the downstream signalling pathways via PI3K.
The chronically exposed cells also showed an induction of the p-AKT/AKT and p-SAPK/JNK pathways, as well as the EMT transcription factors.
- The conclusions should be more clearly delineated. Presenting them in a concise and explicit manner would strengthen the overall impact of the manuscript.
Reply: Yes, we added a conclusion
Chronic kidney disease (CKD) and kidney cancer are both associated with arsenic in drinking water, according to epidemiological studies. Upon acute kidney injury, proximal tubular epithelial cells proliferate rapidly, which is due to their intrinsic proliferation potential. This remains a translational challenge to understand how progenitor cells regenerate after chronic nephrotoxic injury, despite substantial progress in understanding renal fibrosis and the role of progenitor cells. Most studies assume that insults and damage only occur to mature and differentiated, functioning cells and not to regenerating progenitor cells. This is an issue that our system is uniquely positioned to address. An in vitro study carried out by our lab used human renal tubular progenitor cell lines to understand how chronic level of i-As+3 induces EMT-like phenotypic and molecular changes. It is suggested that the concomitant upregulation of MFAP5 and CEACAM 5 and 6 is associated with renal fibrosis and cell transformation. It is anticipated that this study will contribute to the understanding and development of new therapeutic strategies to address RCC caused by chronic exposure to arsenic.
Please ensure that all abbreviations are spelled out in full upon their first appearance in the text. The majority of abbreviations currently lack full names.
Reply: Yes, corrected.
- The Introduction, Materials and Methods, and Discussion sections are particularly well written and informative.
- In the Results section, specifically subsection 3.1, please provide clearer descriptions for P3 and P8. Additionally, there is some repetition in the figure legends for Figures 2 and 3 that should be revised for conciseness. In Figure 6, the Western blot results for phosphorylated JNK are not clearly visible. Enhancing the image quality would help support the findings.
Reply: P3 and P8 refer to the passage number. The passaging of cells in the presence of 4.5 asenite is describe in the Materials and methods. A passage refers to the sequence of seeding the cells into a new flask and continuing culture until the cells grow and fill the flask to confluence, at which time the cells are detached with trypsin and seeded into a new flask for the next passage. As for figure legends 2 and 3, we did not find much redundance, but wording may be similar. We feel we need to specify statistical definitions even though they are similar in other figures. As for Figure 6, we agree that the gel is relatively overexposed. Altering the brightness would enhance the presentation, but we were fearful of altering the image. If the reviewers prefer, we can lighten it in the next revision. It is important, however, to quantify the unaltered image.
Given the quality of the research and the relevance of the topic, I recommend acceptance of this manuscript following minor revisions. The study is well-conducted and will be of significant interest to both clinicians and researchers in nephrology and related fields.

Reviewer 2 Report
Comments and Suggestions for Authors
Dear Authors,
The manuscript “Chronic Exposure of Renal Progenitor Cells (HRTPT) to As (III) 2 Implicates Microfibril Associated Protein 5 (MFAP5) in the 3 Activation of Carcinoembryonic Antigen Related Cell 4Adhesion Molecules (CEACAM 5 & 6)” was reviewed. The authors have tested the acute and chronic exposure of i-As (III) to “Human Renal Tubular 19 Precursor TERT” (HRTPT). The investigators have done a screening for different dosages of i- As(III) on HRTPT and examined the changes in molecular patterns that could potentially contribute to CKD or RCC. The examination of changes to cells on i-As is good and has provided sufficient data to prove the concept; however, some points need to be addressed.
Please address some questions for clarity and enhance the manuscript.
- Why were kidney progenitor cells used and not the kidney cell lines? Since the investigation on the effect of i-As on adult kidney, it is important to use kidney cells to elucidate the mechanism. While using progenitor cells, the impact on signaling that leads to kidney formation can only be studied. How do you justify it? Have you considered that experiment?
- Protein and genes relevant to cancer progression were shown to be regulated in iAs-treated groups. However, cell proliferation or apoptosis markers were not studied. Please include PCNA, Ki67, and apoptosis markers to show cell characteristics.
- There are only 3 replicates. Results are with 6 replicates
- In Figure 6, the legends are described wrongly. Please remove and correct it.
- Figure 7 is mentioned as Figure 6. Please take utmost care while drafting the manuscript.
- In Figure 3, Figure C- Is it the original figure or adapted from elsewhere? If not, have you taken the copyright? It is very important to make original figures.
- In Figure 2 C, it is written R1, 2, and R3 against the control. (What is 2?) Can you explain what that is? The representations should be uniform.
Author Response
Reviewer 2
Please address some questions for clarity and enhance the manuscript.
Why were kidney progenitor cells used and not the kidney cell lines? Since the investigation on the effect of i-As on adult kidney, it is important to use kidney cells to elucidate the mechanism. While using progenitor cells, the impact on signaling that leads to kidney formation can only be studied. How do you justify it? Have you considered that experiment?
Reply: Our in vitro progenitor cell model system is used to assess the effects of arsenite exposure on the process of regenerating the nephron after high levels of cell death. The suggestion of focusing on signal transduction pathways that determine proliferation and differentiation into tubular cells is certainly the ultimate goal, but very little is known about the molecular determinants of tubular differentiation. Currently, we are investigating the effects of nephrotoxicants on the regeneration process and its potential altered outcomes.
Protein and genes relevant to cancer progression were shown to be regulated in iAs-treated groups. However, cell proliferation or apoptosis markers were not studied. Please include PCNA, Ki67, and apoptosis markers to show cell characteristics.
Reply: Cell proliferation markers in culture systems are probably not very useful. Cell proliferation is noted by an increase in cell number. PCNA and Ki67 are usually reserved for tissue sections to show which cells are proliferative in a background of stagnant cells, particularly in the context of tumor histological assessments. Apoptosis markers are useful for investigating the mode of cell death. Our previous research with mortal renal cells and the RPTEC-TERT1 cell line, which contains progenitor populations, does not die by apoptosis when challenged with heavy metals (https://pubmed.ncbi.nlm.nih.gov/15129022/, https://pubmed.ncbi.nlm.nih.gov/28587817/). While apoptosis is still possible in the current experimental system, we feel this is outside the theme of the current study.
There are only 3 replicates. Results are with 6 replicates
Reply I could not understand this question. We have replicates for each passage, P3 we showed three replicates and P8 we had three replicates.
In Figure 6, the legends are described wrongly. Please remove and correct it.
Reply: Legends corrected accordingly to Figure 6
Figure 7 is mentioned as Figure 6. Please take utmost care while drafting the manuscript.
Reply: Yes corrected accordingly to Figure 7
In Figure 3, Figure C- Is it the original figure or adapted from elsewhere? If not, have you taken the copyright? It is very important to make original figures.
Reply: We have given the source reference https://www.proteinatlas.org/ENSG00000105388-CEACAM5/subcellular, there is no copyright issue. This just shows the CEACAM5 is located on the cell surface and act as receptor. I replaced it with my own developed figure via biorender.
In Figure 2 C, it is written R1, 2, and R3 against the control. (What is 2?) Can you explain what that is? The representations should be uniform.
Reply: Yes corrected accordingly, R denotes number of replications.
Round 2
Reviewer 2 Report
Comments and Suggestions for Authors
Authors,
I appreciate you addressing our concerns and making necessary changes. The clarification for the questions is satisfactory and I suggest the manuscript can be accepted for publication.